# New Insights on the Minimal-Invasive Therapy of Cervical Cancer

**DOI:** 10.3390/jcm11164919

**Published:** 2022-08-22

**Authors:** Khayal Gasimli, Lisa Wilhelm, Sven Becker, Rudy Leon De Wilde, Morva Tahmasbi Rad

**Affiliations:** 1Department of Gynecology and Obstetrics, Division of Gynecologic Oncology, Frankfurt Goethe University Hospital, Theodor-Stern-Kai 7, 60590 Frankfurt, Germany; 2Department of Gynecology, Carl-von-Ossietzky University, 26129 Oldenburg, Germany

**Keywords:** cervical cancer, laparoscopy, laparotomy, minimally invasive surgery, oncology

## Abstract

Objective: The ideal management of early-stage cervical cancer has become the subject of a global controversy following the publication of a prospective study in 2018 that reported a worse oncologic outcome when comparing the minimally invasive approach to the laparotomy approach. The discussion involves both prospective and retrospective data and general and theoretical considerations. We wanted to look at the data available today and review the different opinions, offering an impartial assessment of the ongoing controversy. Methods: The available literature was reviewed, focusing on articles arguing for and against minimally invasive surgery in cervical cancer. We tried to avoid any fundamental bias, as is often evident in the available reviews on the subject. Literature both before and after the 2018 publication was taken into consideration. Results: As is usual in discussions of concepts, the literature that is now available provides arguments for both sides of this challenging issue, depending on one’s standpoint. Science-related writing is not immune to trends. There is a curious shift in opinion seen before and after 2018. One must question whether there was a prejudice in favor of minimally invasive surgery prior to the publication of the NEJM articles and a bias against it afterward. Conclusion: Whether further minimally invasive surgery for cervical cancer is invariable is tied to the more pressing question of how this surgery will have to be centralized in the future. Unless these questions are linked, no satisfactory solution can be found.

## 1. Introduction

Cervical cancer remains one of the most common cancers worldwide [1]. Although there is hope that it will be eradicated through HPV-Vaccination, we will not see a significant decline for at least another two to three decades [2]. At the same time, increasingly widespread screening programs will lead to more diagnoses of early-stage and operable cervical cancer [3]. For the next 50 years at least, surgery will remain one of the essential pillars of curative treatment. Minimally invasive surgeries have fundamentally changed abdominal surgery in all aspects. The advantages concerning intra- and perioperative morbidity have consistently been demonstrated [4,5,6]. In 2018 a publication put into question the available scientific evidence up to that point [7]. This has led to an almost ideological discussion about the role of minimally invasive surgery for early-stage cervical cancer [8]. In this article, we will review the available data and arguments and try to offer a rational solution to the current conundrum.

## 2. Data before 2018

Minimally invasive surgery was first introduced into clinical practice in the 1990s [9]. However, it took well into the early 2000s for the technique to become widespread and be used for more advanced gynecologic surgeries [10,11]. The laparoscopic hysterectomy was at the center of the minimally invasive revolution of gynecologic surgery. Once this technique had been mastered, other, more complex surgeries followed: oncologic hysterectomy, laparoscopic lymphonodectomy, laparoscopic management of severe endometriosis, laparoscopic pelvic floor repair, and—finally—radical laparoscopic hysterectomy [5,12,13,14].

After almost 100 years of little technical progress, introducing a completely new surgical approach was an exceptional task. Experienced “open” and “vaginal” surgeons found themselves confronted with a completely new approach and—unusually for a seasoned surgeon—a new learning curve. A vast body of literature proved that even under scientific scrutiny, the minimally invasive approach was not only equal but, in many ways, superior to the traditional approach [15,16,17].

Although today, the adoption of minimally invasive surgeries for gynecologic disease greatly varies between different countries, it is fair to say that modern gynecologic surgery has become mostly minimally invasive [18,19]. Similar fundamental changes have been seen in urologic surgery and general abdominal surgery. However, nowhere has this change been more radical than in gynecology. It is important to reflect on the psychological impact this change has made. The driving force behind a complete overhaul of what constitutes gynecologic surgery within one generation of surgeons was driven by a small group of pioneers and early adopters who relentlessly pushed the limits, by a new, young generation who jumped at this new “modern” and technologically-driven form of surgery and by a tremendous pressure of expectations from the patients [9].

The role of the patients is often overlooked—wrongly so because, as customers within many health systems, they exert tremendous economic power. Medicine and surgery do not evolve along strictly scientific pathways—as is sometimes presumed. How new therapeutic methods become mainstream results from an exciting mix of science, technology, business interests, individual curiosity, personal ambition, and media coverage. Although we hope that science ultimately will refute or back up what we do as doctors—and while we trust that those pathways that are clearly wrong and inferior will be quickly identified by a combination of science and peer review—it is important to remember, that surgical progress is a human process.

In hindsight, the mostly positive retrospective data on the oncologic outcome of minimally invasive cervical cancer surgery probably reflect a publication bias driven by the desire to belong to the “new” and “modern” technological revolution. However, unfortunately, science is not immune to the concept of fashion. During this time, those skeptical voices were often willfully overheard and cast aside as “old-fashioned”.

This was the general context in which the LACC trial was published. Still, in March 2018, the ESGO Guidelines read: “Radical Surgery by a gynecological oncologist is the preferred treatment modality minimal invasive approach is favored” [20].

## 3. The LACC-Trial

The LACC Trial was started in the early 2010s by Prof. Obermaier from Australia [7]. To this day, it remains the only prospective randomized trial on cervical cancer. As the original radical hysterectomy by Ernst Wertheim was introduced long before the age of clinical trials, as were the further modifications by Okabayashi and Meigs, none of the different steps have ever been evaluated in a prospective surgical setting. It is important to remind us that the basis of surgical technique today is mostly what is—often disparagingly—referred to as “Expert Opinion”. In fact, the introduction of minimally invasive techniques served as the starting point to scientifically assess the true value of novel surgical techniques—regarding morbidity, complications, and outcome.

The LACC trial was conducted in the spirit of this. As portrayed by the main author, the results of the LACC trial were perfectly clear.

However, both the absolutism and the forcefulness with which these claims have been made have created a critical backlash, highlighting important weaknesses of the study:

The study had a slow uptake, causing the recruiting timeframe to be nine years (June 2008–June 2017).

Although 33 centers participated, the contribution was very heterogenous—and has never been published in detail. Mathematically, when 33 centers recruit 631 patients over 9 years, this means that each center recruited—on average—19.12 patients over the course of the study, leading to the recruitment of 2.1 patients per center per year. The questions raised by such a simple mathematical analysis are apparent and have never been addressed.

However, it is easy to criticize a prospective randomized surgical study. Contrary to medical trials in oncology, surgical trials do not lend themselves easily to what has become the gold standard of scientific inquiry: The prospective, randomized, blinded, multicenter trial. The most important factor regarding the medical quality, the surgeon, cannot be randomized. One surgeon can unlikely perform two different surgical techniques with equal authority. A “blinding” of the process is difficult.

The party that does not like the outcome can easily pick on methodologic deficiencies. One good example of this is the controversy surrounding the prospective trials for neoadjuvant treatment of ovarian cancer. Prospective surgical trials are always imperfect. They reflect not an ideal world but the messy reality of daily surgical routine. As such, the quality of the LACC trial, even where it is deficient—must be commended. Thus, drawing attention to critical aspects of the trial does not question the validity of the results. I merely try to provide perspective. Already at ASCO 2018, some of these critical points were summarized (Figure 1).

The most important quality control was a video sent in to show surgical qualifications. Although there is no doubt that this led to a high baseline quality—as seen in the excellent survival data in both arms, it ignores the fact that, globally speaking, this trial was still conducted during the “learning curve” period.

The most important criticism focuses on an amazing aspect: The progression-free survival and the overall survival in the minimally invasive group were not bad. In fact, it ended up exactly where most other trials also reported their PFS and OS values. What was remarkable was an excellent good PFS/OS in the open group, with values approaching 98%. (Table 1) [21,22,23]. Thus, the main statement of the data presented was that while the minimally invasive approach was as good as any historical data available for comparison, the PFS/OS values for open surgery under controlled circumstances were much better than previously thought (Table 2) [24,25,26,27].

In fact, oncologic outcomes similar to the LACC-Trial were seen only in single center, single (expert) surgeon publications.

## 4. Data after 2018

Given the overwhelmingly positive retrospective data on the oncologic outcome of minimally invasive cervical cancer surgery published up until 2018, the results of the LACC trial led to a curious change in the publication patterns. In the years following 2018 retrospective results that appeared to support the negative view now taken on laparoscopic or robotic surgery have been published [28,29,30]. It is hard to understand this shift with scientific arguments alone fully. As the overall discussion starts around 2019, publications become more balanced. Currently, by either ignoring or highlighting “Pro” and “Con” papers, the reviewer could easily convey the picture of minimally invasive surgery as oncologically acceptable or as completely unacceptable. The discussion has become a controversy with ideologic undertones. That does not mean that the papers are scientifically not valuable. One underlying theme of the available publication is the level of centralization that becomes transparent. The most convincing retrospective data comes from the Scandinavian countries (Sweden and Denmark, mostly), where oncologic surgery in general and cervical cancer surgery, in particular, has been highly centralized by law. The Scandinavian centers performed a comprehensive analysis of their data and could not find any difference in the oncologic outcome.

One publication from the UK [31] compiled the data of eight UK gynecology centers with proven expertise in cervical cancer surgery, including minimally invasive techniques. A total of 779 cases were reviewed, of which 78% were done by either the laparoscopic or the robotic approach. After a median follow-up of 23 months. With only 36/779 recurrences (4.6%), there was no mortality difference between the two groups. The preliminary data from the Swedish National Registry looks at the data from 2011 to 2017. A total of 852 cases from six sites (3 surgeons/site) showed no difference in disease-free survival (overall survival was 92%).

In 2020, the Danish group reported on their nationwide survival after adopting robotic minimally invasive surgery for early-stage cervical cancer [32]. In this study, coming from a country with highly centralized cervical cancer care, no difference was seen between the different groups, and no disadvantage was discerned for the minimally invasive group. 

Of particular interest, given the (almost) complete abandonment of minimally invasive radical hysterectomy in the United States for legal reasons specific to this country, is the retrospective series from Memorial Sloan Kettering Hospital [33]. Looking at 196 evaluable cases, of which 117 were minimally invasive (106 robotic, i.e., 90%) and 79 were done by laparotomy, do differences in oncologic outcome were seen in two cohorts that were similar for age, BMI, substage, histology, and clinical and pathologic tumor details. 

In this context, it must be stated that most surgical centers in Germany and southern Europe perform less than 20 radical hysterectomies per year. In fact, most centers perform less than 10 per year. 

In view of the available literature on learning curves [33,34], it is unlikely that with less than 20 radical hysterectomies, any adequate teaching can be performed. Existing competencies might be preserved. However, even the maintenance of adequate practice would seem doubtful. The main authors of the LACC trial must be commended for their honesty, as the surgical data from MD Anderson was made available by them and shows that the true problem of cervical cancer surgery is probably not whether the surgery is performed open or by minimally invasive technique, but whether a center as sufficient numbers at all, to offer an adequate surgical routine.

## 5. Theoretical Considerations

Although the main argument of the LACC trial was not a bad oncologic outcome of the minimally invasive approach but a superb oncologic outcome of the open approach, there have been various attempts to explain the difference:(1)The CO_2_-Theory.

The gas used to create the necessary pneumoperitoneum has been looked at many times for its oncologic side effects. Does CO_2_-gas stimulate oncologic growth? Or is the insufflation and subsequent “ex”-sufflation a merely mechanical way to distribute cancer cells more easily inside the abdominal cavity? The existing literature is equivocal [35].

(2)The most interesting critical point has been the theory that during a laparoscopic (or robotic) procedure, it is not possible to completely “isolate” the tumor mass from the abdominal cavity, allowing for the mechanical “entry” of cancer cells into the peritoneal cavity.

This has led to corrective techniques adapted from the Shauta radical vaginal hysterectomy that attempts to dissect a vaginal cuff, which is then used to “cover“ the exposed tumor [36]. Preliminary results have been quite encouraging.

Most of the available literature, including the LACC-trial results, see a distinct advantage for Cervical Cancer tumors larger than 2 cm in the open group. Larger cervical tumors tend to be necrotic and have an obvious and clinically apparent tumor cell shedding. The idea that intraperitoneal shedding of a tumor which is usually strictly extraperitoneal does make immediate sense. However, it raises the question of whether malignancy can be understood in mechanical terms. For example, endometrial cancer cells are routinely spread into the abdominal cavity during hysteroscopy. Breast cancer cells are routinely “spread” into the bloodstream during true-cut biopsy.

(3)One of the most significant points of discussion is whether the cutoff of two centimeters is real. Again, different publications tell different stories. Although the original publications of the LACC trial and the epidemiological study of Alexander Melamed MD, MPH clearly see a cutoff at 2 cm and further studies from other centers seem to confirm this [37,38,39], other retrospective studies have failed to find such a limit [29,40].

## 6. Moral Considerations

Although the available publications focus on oncologic outcomes and technical details, one important aspect regarding the underlying issues is too often overlooked. It is generally accepted that science continually advances. Scientific progress is taken for granted. New treatments are expected to replace old ones. New diagnostic techniques are expected to improve on old ones.

Implicitly, we expect the field of surgery to develop similarly. Surgery, however, is a learned trade that involves an individual surgeon with individual skill and experience, both of which inevitably are subject to a personal history of training and experience. Particularly the process of teaching surgery is often taken for granted, even though it involves an arduous year-long process. When “new” techniques are adopted, how are they to be introduced when the “experimentation” on patients is generally considered unacceptable? There is no clear answer to this, except that the generally high standard of care in most of the developed world seems to guarantee a basic safety net that avoids the most egregious pitfalls of a process that cannot avoid using a new and yet unproven technique on individual patients in order to turn it into a routine and proven one.

## 7. Ongoing Studies

The LACC-Trial took nine years to reach sufficient maturity. Even then, strictly speaking, its statistical aim was not reached. New prospective trials both for laparoscopy and robotic surgery are currently recruiting. Realistically, no new results are to be expected before 2025. Until then, many surgeons have stopped using the minimally invasive approach except for small-size tumors, ideally after complete removal by conization (with clear margins). Whether or not this is a rational decision will never be known, as surgeons trained and experienced in the minimally invasive technique now switch back to a technique, they were much less proficient in. One of many not openly discussed aspects of the ongoing discussion. China has long been wary of American and European data, considering that the sheer numbers of Chinese oncologic centers far exceed those of comparable US or European centers. Chao X. et al. [41] announced a Phase III multicenter randomized controlled trial in the British Medical Journal Open in 2019.

Of particular interest will be the results of the RACC Trial [42]. Most of the data supporting minimally invasive surgery has been robotic, while most of the minimally invasive surgery of the LACC trial was laparoscopic. Therefore, the RACC Trial (Robot-assisted Approach to Cervical Cancer) will focus on this MIS technique only. One other ongoing trial that could potentially affect the way we look at the LACC-trial results is the SHAPE trial [43]. The SHAPE trial looks at a possible reduction in radicality in early-stage cervical cancer without risk factors. Should the SHAPE trial show, that simple hysterectomy might be adequate for stage IA cervical cancer or even some stage Ib1, the question arises, whether laparotomy must then be done for a simple hysterectomy.

## 8. Outlook

Minimally invasive surgery has revolutionized gynecologic surgery, including many aspects of gynecologic oncology. Coming of age during the time of rigorous scientific evaluation, no new surgical techniques have been more thoroughly evaluated, as exemplified by the prospective trials for endometrial cancer treatment. The LACC trial must be seen in this context as one more step towards improving the quality we provide to our patients. However, no one trial can be the end of scientific inquiry. The LACC trial forced us to reflect on the quality and relevance of what we do. Surgical training in an age of increasing specialization, a lack of qualified surgeons, and ever-increasing economic pressure pose considerable challenges to the current and future generation of (gynecologic) surgeons.

## Figures and Tables

**Figure 1 jcm-11-04919-f001:**
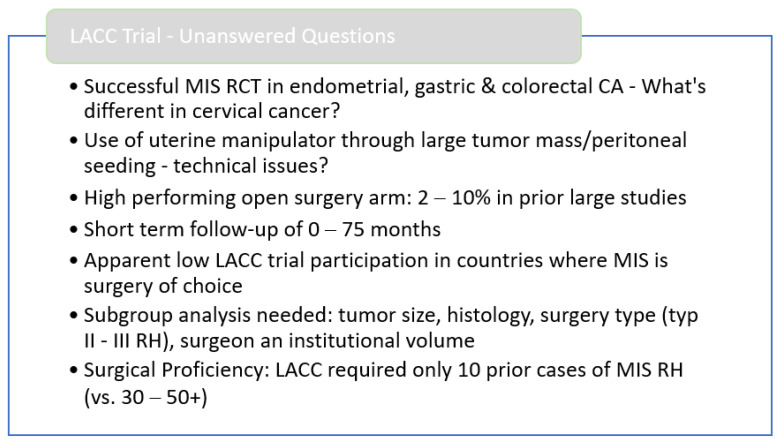
Summary of initial criticism at ASCO 2018 presented by Ginger J. Gardner, MD.

**Table 1 jcm-11-04919-t001:** Comparison of recurrence rates (RR) in different surgical trials of cervical cancer [21,22,23].

Trial	Open			MIS/Robot		
	N	Recurrences	RR (%)	N	Recurrences	RR (%)
LACC 2018	312	7	2.2	319	24	7.5
Shiah et al., 2017	202	21	10.4	109	11	10.1
Wallin et al., 2017	155	16	10.3	149	20	13.4
Sert et al., 2016	232	21	9.0	259	23	9.0
Zanagnolo et al., 2016	104	11	10.6	203	18	8.8

**Table 2 jcm-11-04919-t002:** Comparison of oncologic outcome in expert surgeon trials [24,25,26,27].

Laparotomy		Laparoscopy	
Cibula et al., 2011	Hockel et al., 2009	Nie et al., 2017	Chiantera et al., 2016
IA-IIB	IB-IIB	IA1-IIA2	IA2-IB1
Radical hysterectomies	TMMR	Robot vs Laparoscopic RH	L-TMMR
Median FU 55 months	Median FU 41 months	Median FU: not reported	Median FU 18 months
120 IB1	159 IB1–IIA (total 212)	IB1 592	IB1: 61
Adjuvant treatment 6.4%	Adjuvant treatment 3.7% (No RT)	Adjuvant treatment 40% robotic and 55% laparoscopic	Adjuvant treatment 36.6%
Recurrences in IB1: 3/120 (2.5%)	Recurrences 3/159 (1.9%)	Recurrences in 32/856 (3.7%) (0 recurrences in robotic arm)	Recurrences 2/71 (2.8%)

## Data Availability

Not applicable.

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
