# Peer review of "New Insights on the Minimal-Invasive Therapy of Cervical Cancer"

_jcm, 2022, doi:10.3390/jcm11164919_

Round 1

Reviewer 1 Report

This article reminded us that the weakness of the LACC trial designs should be re-evaluated. Reproducibility of the open group result in LACC trial might be reevaluated. Also, MIS data from center with sufficient case number and fair modality should be emphasized. 

The pointed problems and designs errors were fairly though-provoking and persuasive. However, there're lots of English grammatical, capitalization and punctuation marks errors needs to be refined in the article.

Author Response

dear Reviewer,

thank you very much for your favorable reviews. We greatly appreciated your valuable input. We took all your recommendations into consideration and added your references as requested. Furthermore, we appreciate your thoughts on the controversy surrounding the LACC trial.

Again, thank you very much for your efforts, we look forward to continuing this highly relevant discussion for the benefit of all of our patients.

With the most sincere and most grateful greetings,

Dr. Khayal Gasimli, for all the authors.

Reviewer 2 Report

The paper is very interesting with several approaches concerning different considerations.

LACC trial is a cornerstone of the article, for discussion, as the main study which open different questions that authors tried to put in one place.

The authors have to notice that even we have conflicting results, in light of the findings of the LACC trial, ESGO and NCCN changed their recommendations in favor of laparotomy in the surgical management of cervical cancer. (ref. 1.National Comprehensive Cancer Network. NCCN Clinical Practice Guidelines in Oncology Cervical Cancer. 2022. Available online: https://www.nccn.org/professionals/physician_gls/pdf/cervical.pdf (accessed on 4 February 2022). 2.Querleu, D.; Cibula, D.; Concin, N.; Fagotti, A.; Ferrero, A.; Fotopoulou, C.; Knapp, P.; Kurdiani, D.; Lederrmann, J.A.;  Mirza, M.R.; et al. Laparoscopic radical hysterectomy: A European Society of Gynaecological Oncology (ESGO) statement. Int. J. Gynecol. Cancer 2020, 30, 15. [CrossRef] [PubMed])

 The authors point very well that the results of LACC opening unanswered question that was explored some of the potential causes that tried to explain the poorer oncologic outcomes associated with MIS, including the type of MIS surgery, the size of the lesion, the impact of CO2 pneumoperitoneum, prior conization, the use of uterine manipulator, the use of protective measures, and the effect of surgical expertise/learning curve.

Suggest the authors to use for references:

Ref. Touhami O, Plante M. Minimally Invasive Surgery for Cervical Cancer in Light of the LACC Trial: What Have We Learned? Curr Oncol. 2022 Feb 14;29(2):1093-1106. doi: 10.3390/curroncol29020093. PMID: 35200592; PMCID: PMC8871281.

 Concerning that facts authors very well try to explain the problems that could be for the future to reduce MIS in treatment of cervical cancer but even that I have to point that early cervical cancer tumor less than 2 cm is on open area with question will it be enough simple hysterectomy with SLN detection but as MIS approach. We are waiting the results of SHAPE study or CONTESA in fertility spearing approach.

Also nowdays in era of primary and secondary prevention, less radicality for small tumors as option maybe laparoscopy have to put in different context  not for radical hysterectomy but for TLH with SLN detection or even lymphadenectomy and fertility spearing approach so we are not losing MIS as way of treatment of cervical cancer. I will try to put it in that context.

I do think that MIS have place in cervical cancer in different context that will be more in future.

So by decreasing the incidence of cervical cancer by using the vaccination and screening, will we have by centers more than 20 operations radically by surgeons for good learning curve even in open approach.

As a reviewer of this article, which is the great honor for me,  just question do we spend our time to try to explain something that will be maybe no more option in the future just in some cases that we need centralization of course. And in countries where incidence of cervical cancer is high do we even have robotic surgery.

The literature is well balanced.

Author Response

Dear Reviewer,

thank you very much for your favorable reviews. We greatly appreciated your valuable input. We took all your recommendations into consideration and added your references as requested. Furthermore, we appreciate your thoughts on the controversy surrounding the LACC trial.

Again, thank you very much for your efforts, we look forward to continuing this highly relevant discussion for the benefit of all of our patients.

With the most sincere and most grateful greetings,

Dr. Khayal Gasimli, for all the authors.

Reviewer 3 Report

The topic covered is of high relevance and will be interesting to readers. There have been a few reviews about MIS in cervical cancer in recent years. This one is original in a way that presents reasons for the paradigm shift in cervical cancer surgery that are not derived only from scientific conclusions, but from a mix of science, fashions in publishing, and the constant human will to change and improve.  Paragraphs that cover surgical competence, learning curves, ethical considerations, and randomized trials in surgery are extraordinarily written. 

References in Section 4. should be added in a place where " retrospective results that appeared to support the negative view on MIS" are mentioned. 

A minor spell check is required. 

In general, the article is well written. 

Author Response

(The authors gave the same response as above.)
